# Clinical Characteristics and Management of Patients with a Suspected COVID-19 Infection in Emergency Departments: A European Retrospective Multicenter Study

**DOI:** 10.3390/jpm12122085

**Published:** 2022-12-19

**Authors:** Anthony Chauvin, Anna Slagman, Effie Polyzogopoulou, Lars Petter Bjørnsen, Visnja Nesek Adam, Ari Palomäki, Andrea Fabbri, Said Laribi

**Affiliations:** 1Emergency Department and PreHospital EMS, Lariboisiere Hospital, Assistance Publique Hôpitaux de Paris, 75610 Paris, France; 2Inserm U942 MASCOT, University of Paris, 75015 Paris, France; 3Departments of Emergency and Acute Medicine, Campus Mitte, Virchow-Klinikum Charité-Universitätsmedizin, 10117 Berlin, Germany; 4Emergency Medicine Department, Attikon University Hospital, 12462 Athens, Greece; 5Clinic of Emergency Medicine and Prehospital Care, Department of Circulation and Medical Imaging, St. Olav’s University Hospital, Norwegian University of Science and Technology (NTNU), 7034 Trondheim, Norway; 6Resuscitation and Intensive Care, University Department of Anesthesiology, Sveti Duh, University Hospital, 10000 Zagreb, Croatia; 7Emergency Department, Division of Medicine, Kanta-Häme Central Hospital, 13530 Hämeenlinna, Finland; 8Faculty of Medicine and Health Technology, Tampere University, 33100 Tampere, Finland; 9Emergency Department, Presidio Ospedaliero Morgagni-Pierantoni, AUSL Romagna, 47121 Forlì, Italy; 10Emergency Medicine Department, Tours University Hospital, 37044 Tours, France; 11School of Medicine, Tours University, 37000 Tours, France

**Keywords:** severe acute respiratory syndrome coronavirus 2 (SARSCoV-2), emergencies, COVID-19, dyspnea

## Abstract

Background: Our aim is to describe and compare the profile and outcome of patients attending the ED with a confirmed COVID-19 infection with patients with a suspected COVID-19 infection. Methods: We conducted a multicentric retrospective study including adults who were seen in 21 European emergency departments (ED) with suspected COVID-19 between 9 March and 8 April 2020. Patients with either a clinical suspicion of COVID-19 or confirmed COVID-19, detected using either a RT-PCR or a chest CT scan, formed the C+ group. Patients with non-confirmed COVID-19 (C− group) were defined as patients with a clinical presentation in the ED suggestive of COVID-19, but if tests were performed, they showed a negative RT-PCR and/or a negative chest CT scan. Results: A total of 7432 patients were included in the analysis: 1764 (23.7%) in the C+ group and 5668 (76.3%) in the C− group. The population was older (63.8 y.o. ±17.5 vs. 51.8 y.o. +/− 21.1, *p* < 0.01), with more males (54.6% vs. 46.1%, *p* < 0.01) in the C+ group. Patients in the C+ group had more chronic diseases. Half of the patients (*n* = 998, 56.6%) in the C+ group needed oxygen, compared to only 15% in the C− group (*n* = 877). Two-thirds of patients from the C+ group were hospitalized in ward (*n* = 1128, 63.9%), whereas two-thirds of patients in the C− group were discharged after their ED visit (*n* = 3883, 68.5%). Conclusion: Our study was the first in Europe to examine the emergency department’s perspective on the management of patients with a suspected COVID-19 infection. We showed an overall more critical clinical situation group of patients with a confirmed COVID-19 infection.

## 1. Introduction

Coronavirus disease 2019 (COVID-19) was initially reported in Wuhan, Hubei Province, China, in December, 2019, and rapidly spread to all other provinces of China and throughout the world [1,2]. The outbreak of severe acute respiratory syndrome coronavirus 2 (SARS-CoV-2) has a large spectrum of clinical presentations, from the absence of symptoms to the most severe acute respiratory failure associated with high death rates [3]. On 11 March 2020, the World Health Organization (WHO) declared the outbreak a pandemic and stated that Europe had become the epicenter of the pandemic [4].

Over the course of the pandemic, the management of COVID-19 patients has changed drastically. At the beginning of the pandemic, most suspected COVID-19 cases were hospitalized, whereas now only severe patients, namely oxygen-requiring patients, are hospitalized. As a result, in some regions, during their initial evaluation in the emergency department (ED), eligible patients have been offered ambulatory care with monitoring using a dedicated platform (i.e., COVIDOM) [5]. However, the patient’s condition can deteriorate and thus patients may need to be hospitalized in a ward or an intensive care unit [6].

For the time being, uncertainty remains in the management of these patients. Although evidence relating to the death and adverse outcomes of COVID-19 is rapidly accumulating, most studies focus on the comparison of clinical characteristics between deceased and recovered patients [7,8,9]. Some researchers have explored prognostic factors; however, data have often been monocentric, with relatively small sample sizes, using univariate analysis alone with a lack of clear clinical outcomes for all patients [10,11,12,13]. Moreover, molecular assays (RT-PCR) are considered the reference standard for COVID-19 diagnosis [14], while, when performed on the nasopharyngeal swab samples, this assay could be falsely negative, with up to 30% of patients with clinically and radiologically suspected COVID-19 [15,16]. Few studies are published on the ED management of patients presenting with a suspected COVID-19 infection.

In this study, we aimed to describe and compare the patient profile and outcome of patients attending the ED with a confirmed COVID-19 infection with patients with a suspected COVID-19 infection without a biological or radiological confirmation.

## 2. Methods

### 2.1. Design

This is a European multicenter retrospective study in 21 EDs in seven European countries: Croatia (*n* = 1), Finland (*n* = 2), France (*n* = 9), Germany (*n* = 2), Greece (*n* = 3), Italy (*n* = 3), and Norway (*n* = 1).

### 2.2. Patients and Data

We included all patients who received consultation in participating EDs and who attended the ED with a suspected COVID-19 infection. The period of inclusion was between 9 March and 8 April 2020.

The group with a confirmed COVID-19 diagnosis (C+ group) was retained in symptomatic patients if they had at least one positive diagnostic test (i.e., molecular assay (RT-PCR) or chest CT scan) at the initial consultation in the ED [17,18]. Patients who were PCR-positive but asymptomatic were not included in the study. The chest CT scans were interpreted by radiologists at each site. A chest CT scan was defined as compatible with the diagnosis of SARS-CoV2 if it included ground-glass opacity, consolidation, reticulation/thickened interlobular septa, or nodules [19,20].

The group of patients with a clinical suspicion of COVID-19 and a non-confirmed COVID-19 infection (C− group) were defined as patients with a clinical presentation in ED suggestive of COVID-19, but if tests were performed, they showed a negative RT-PCR and/or a chest CT scan not in favor of COVID-19 diagnosis. We excluded from the analysis patients with a suspicion of COVID-19, for whom either RT-PCR or chest CT scan was not performed.

Study data were obtained from the ED’s patient chart review for each center by local study investigators. For hospitalized patients, local investigators checked the hospitalization report to evaluate the status of patients at 30 days.

For each patient analyzed, a local investigator collected data according to a standardized case report form: (1) patient demographics (i.e., age, sex, and medical history), (2) history of COVID-19 contamination (i.e., healthcare worker, institutional living, COVID-19 contact), (3) clinical signs suggestive of COVID-19, (4) vital parameters at ED arrival (temperature, heart ratio, respiratory rate, blood pressure, oxygen saturation, and mental status according to the Glasgow Coma Scale), (5) laboratory test results, (6) ED treatment (i.e., oxygen therapy, antibiotics), and (7) disposition after ED management. For the 30-day period following the initial ED consultation, a local investigator checked the local electronic health system to see if the patient had either revisited the ED, had been hospitalized within the 30 days, or if death had been reported. There was no follow-up with the recall of patients.

### 2.3. Objectives

The primary objective of this study was to describe and compare the profiles and outcomes of patients attending the ED with a confirmed COVID-19 infection with patients with a suspected COVID-19 infection in a European patient population. Patients with a confirmed COVID-19 infection were compared with patients with a negative COVID-19 test.

### 2.4. Ethics

This study was performed in accordance with the Declaration of Helsinki. Ethics committee approvals were obtained for all participating sites according to local requirements. The population of interest for this study was patients presenting to an ED with suspected COVID-19.

### 2.5. Data Analysis

The statistical analysis was performed using SAS 9.3 (SAS Inst. Inc., Cary, NC, USA). Baseline characteristics were expressed as a number (%) for categorical variables and a mean (standard deviation (SD)) or median (interquartile range (IQR)) for continuous variables, depending on their distribution. Chi-square, Student, and Kruskal–Wallis tests were used for univariate analysis, and logistic regression was used for multivariate analysis and subgroup analysis, estimating odds ratios (ORs) and 95% confidence intervals (CIs).

Differences between groups were compared using Chi-square analysis for qualitative variables and *t*-test for quantitative variables. A *p*-value < 0.05 was considered statistically significant.

## 3. Results

A total of 7876 patients were recruited in this study, 112 (1.4%) of whom were excluded because the proportion of available data for these patients was below 10%. A total of 332 (4.2%) other patients were excluded from the analysis because they did not have RT-PCR or chest CT scans.

Finally, we analyzed 7432 patients in two groups according to their COVID-19 status: the C+ group with a confirmed COVID-19 infection (*n* = 1764, 23.7%), and the C− group with a non-confirmed COVID-19 infection (*n* = 5668, 76.3%) (Figure 1).

The RT-PCR was performed in almost all the patients (*n* = 1645, 93.3%) in the C+ group and only in three-quarters of cases in the C− group (*n* = 4262, 75.2%) (Table 1). The positivity of the RT-PCR was 82.6% *n* = 1359) in the C+ group. Chest radiography was performed in 33.7% and 35.5%, respectively. It was interpreted as normal in 16.4% of the C+ group and 38.3% of the C− group (*p* < 0.01). The chest computed tomography (CT) scan was performed on 1192 patients (67.6%) in the C+ group, whereas only 1406 patients (24.8%) in the C− group had a CT scan. The majority of patients in the C+ group featured thoracic lesions on the CT scan in favor of a COVID-19 diagnosis (Table 1).

Details of the C+ group were as follows: 715 (40.5%) patients with positive CT scans and RT-PCRs, 572 (32.4%) patients with positive RT-PCRs without CT scans performed, 286 (16.2%) patients with positive CT scans but negative RT-PCRs, 119 (6.7%) patients with positive CT scans but without RT-PCR performed, and 72 (4.2%) patients with positive RT-PCRs but negative CT scans.

### 3.1. Demographic Description

The population was older (63.8 y.o. +/− 17.5 vs. 51.8 y.o. +/− 21.1, *p* < 0.01), with more males (54.6% vs. 46.1%) in the C+ group than in the C− group (*p* < 0.01). Moreover, patients in the C+ group had more than patients in the C− group (i.e., diabetes mellitus all types, arterial hypertension, chronic heart failure, coronary artery disease, or stroke).

The proportion of patients with factors of immunosuppression was found to be lower in the C+ group (5.8% vs. 6.8%). Active smoking and chronic alcoholism were less represented in the C+ group (11.2% versus 21.2% for smoking and 4.2% versus 6.5% for chronic alcoholism, respectively).

The details of demographic characteristics are presented in Table 2.

### 3.2. COVID-19 Symptoms and Possible Mode of Contamination

The delay between the symptom onset and the ED arrival time was 6.6 +/− 4.7 days in the C+ group and 5.4 +/− 5.6 days in the C− group (<0.01). The proportion of institutional living and patients who had come into contact with a COVID-19 patient was higher in the C+ group.

The three most frequent symptoms were the same in both groups but were not found at the same frequency, namely, self-reported feverishness (68.9% vs. 41.2, *p* < 0.01), cough (67.4% vs. 61.4%, *p* < 0.01), and shortness of breath (55.3% vs. 44.0%, *p* < 0.01). Anosmia was described in 8.8% (*n* = 153) in the C+ group and 2.9% (*n* = 163) (*p* < 0.01) in the C− group (Table 3).

### 3.3. Vital Parameters and Clinical Examination at ED Arrival

Patients in the C+ group appeared to be in a more critical clinical situation than patients in the C− group. Indeed, the proportion of patients with an altered mental status was higher in the C+ group (4.1% vs. 2.1%, *p* < 0.01). Similarly, clinical signs of respiratory distress were more frequent in patients in the C+ group. The most frequent sign of respiratory distress was supraclavicular pulling (8.6% in the C+ group vs. 3.4% in the C− group). However, the proportion of patients with dyspnea was less in the C+ group (41.6% vs. 53.8%). The pulmonary auscultation was normal in one-third of patients in the C+ group, but in two-thirds of patients in the C− group (38.4% vs. 77.7%, *p* < 0.01) (Table 4).

### 3.4. Tests Performed at ED Arrival

The mean white blood count was lower in the C+ group than in the C− group (8.6 +/− 9.3 versus 10.2 +/− 8.4). However, biomarkers were significantly higher in patients with a confirmed COVID-19 infection. Indeed, the mean levels of the D-dimer and the C-Reactive Protein (CRP) were 1072 +/− 1057 and 86.2 +/− 80.4 in the C+ group, and 756 +/− 968 and 45.4 +/− 70.4 in the C− group, respectively. The mean level of lactate was higher in patients with a confirmed COVID-19 infection (1.3 +/− 0.83 vs. 0.9 +/− 1, *p* < 0.01). The mean level of procalcitonin was not statistically different between both groups (*p* = 0.09), as seen in Table 5.

### 3.5. ED Therapeutic Management

Half of the patients (*n* = 998, 56.6%) in the C+ group needed oxygen compared to only 15% in the C− group (*n* = 877). The use of non-invasive ventilation in the ED was higher in the C+ group in comparison with the C− group, at 14.2% (*n* = 142) vs. 5.4% (*n* = 47), respectively (*p* < 0.01).

Antibiotics were prescribed to 20.0% (*n* = 350) and 15.8% (*n* = 896), while antivirals were used for 2.2% (*n* = 39) and 0.6% (*n* = 32) in both the C+ group and C− group, respectively (for both *p* < 0.01) (Table 6).

### 3.6. Patient Outcomes after ED Management

Two-thirds of patients from the C+ group were hospitalized in the ward (COVID-19 unit) (*n* = 1128, 63.9%), whereas two-thirds of patients in the C− group were discharged after their ED visit (*n* = 3883, 68.5%). Patients in the C+ group who were discharged from the ED returned to the ED more often (15.8% vs. 8.3%, *p* < 0.01), were more often hospitalized (12.3% vs. 2.7%), and had a higher mortality rate (1.1 vs. 0.4, *p* = 0.02) during the 30-day follow-up period when compared with patients in the C− group who were discharged from the ED. Among the 4338 patients discharged from the ED, 653 (15%) had a CT chest scan in the ED. Around half of the hospitalized patients had a chest CT scan (*n* = 1754/3038, 57.7%).

Direct ICU hospitalization after ED medical care was more frequent in the C+ group than in the C− group (9.5% vs. 3.3%). Details of the patient outcomes are presented in Table 7.

## 4. Discussion

Our study provided the clinical characteristics and outcomes of patients with confirmed or suspected COVID-19 in 21 ED from 7 European countries. This study provided an additional overview of the patient characteristics, treatment, and outcomes of COVID-19 in EDs.

Our study was pragmatic because we included all patients with a suspicion of COVID-19 infection and not just patients with a confirmed COVID-19 infection based on RT-PCR. In fact, during the first phase of the pandemic, the strategies for performing RT-PCR in EDs varied from one country to another, and also between centers, depending on the availability of RT-PCR tests. In France, for example, the strategy evolved from performing an RT-PCR only for hospitalized patients to performing the test on all patients suspected of having COVID-19; as a result, different test execution strategies could inevitably introduce a risk of bias. Furthermore, it has been shown that an RT-PCR performed early can be negative [21]. In our cohort, the delay in consultation of patients in the C− group was 5.4 +/− 5.6 days, which confirms the early consultation and therefore a possible cause of RT-PCR negativity. RT-PCR is considered to be the gold standard in the diagnosis of COVID-19. However, with a sensitivity of about 70%, this approach is questionable. Moreover, in the first wave, the decline in the overall number of patients in the ED showed that the majority of ED visits for dyspnea were related to COVID-19. During the first wave of the COVID-19 pandemic, the number of usual patient visits to the ED reduced. The probability of COVID-19 was reinforced because influenza or other respiratory viruses were rarely diagnosed during those weeks [22]. During the period of high COVID-19 prevalence, Peyrony et al. reported that the RT-PCR result was more likely to be negative when the emergency physician thought that the clinical probability was low, and more likely to be positive when they thought that it was high [23].

The place of the chest CT scan in patient management needs to be discussed, since only around 15% of the non-hospitalized patients had a chest CT scan, whereas 57.7% of the hospitalized patients had one. Many studies have evaluated the relationship between the severity of lung damage on chest CT and mortality [24]. It would appear that lung damage alone is not a factor associated with mortality [25]. Despite this, it seems legitimate, due to the thrombo-embolic risk of COVID-19, to perform a chest CT scan with injection [26]. Concerning ambulatory patients, the place of the CT scan, and in particular the injection of contrast products, remains unclear. The disease in these patients was less severe and therefore did not require systematic biological investigations, which could have led to a chest CT scan (i.e., positive D-Dimer dosage). If the equipment for the use of the ED is limited, it seems essential to guard against over-testing and to therefore limit access to outpatients only in specific situations (i.e., suspicion of pulmonary embolism) during a pandemic, where access to resources such as chest CT scan can be complicated [27]. The overuse of the CT scan in the first wave may be directly linked to an organizational problem. Indeed, the delay for the RT-PCR result was of the order of 24 h, so the scanner was used as a patient triage tool [28].

Among 4338 patients discharged from the ED, we reported that 396 patients (9.1%) returned to the ED after the initial assessment. In particular, the proportion of readmissions was twice as high in the C+ group (*n* = 72/455, 15.8%) than in the C− group (*n* = 324/3883, 8.3%). This should be seen in the context of the secondary deterioration of COVID-19 patients after a few days of presenting symptoms [29]. Many predictive scores have been developed and validated to identify a subgroup of COVID-19 patients with a low risk of adverse outcomes who can be treated at home safely [30]. We did not study the frequency of readmissions over time. However, we may assume that with the increase in knowledge of COVID-19’s pathology and the increase in hospital capacities, it is possible that this proportion was not stable and that the rate of revisits has decreased over time. To explore this hypothesis, a longitudinal follow-up study is needed.

About one fifth of the patients in the C+ group received antibiotics. However, that treatment option has not yet been shown to be of any benefit in terms of patient survival [31]. This may be explained by the fact that, prior to the emergence of corticoids, antibiotics were the only treatments available to emergency physicians in the treatment of this pulmonary infection [31]. The fact that 39 patients received antivirals is probably related to the initial hydroxychloroquine controversy [32]. It is interesting to note that only half of the C+ group patients needed oxygen therapy (*n* = 998, 56.6%). Indeed, due to the high rate of hospitalization (64%, *n* = 1128) and the specific lung involvement of COVID-19, we would have expected a higher proportion than that observed in the patients requiring ventilatory support. One explanation may be that frail patients or patients with many comorbidities are hospitalized. Exploring the management strategy for these patients could shed some interesting light. The more frequent use of non-invasive ventilation than invasive ventilation for COVID-19 patients (14.2% versus 4.4%, *p* < 0.01) follows research into the lack of superiority in early intubation for COVID-19 patients [33]. This difference in ventilatory mode is not found in the C−group (5.4% versus 5.8%).

Our study had several limitations. First, we performed a retrospective chart review study. However, the loss of data is plausibly limited because of the reliability of electronic medical records in relation to the standardized writing of COVID-19 patient records and laboratory information systems. Second, we did not prospectively follow up with the included patients, and we did not collect the results of RT-PCRs performed after ED discharge. Therefore, some patients could have false negatives with a PCR test that would be positive at a later stage. However, our study was intended to be practical in that it provided an overview of the ED management of COVID-19 patients. Moreover, we only analyzed the data of the first wave in 2020. Since then, epidemiology and management have evolved over time with experience, improved screening techniques, and different COVID-19 variants. Therefore, the generalization of the results may be questionable.

## 5. Conclusions

Our study was the first in Europe to examine the emergency physician’s perspective on the management of patients with a suspected COVID-19 infection. Overall, we found a more critical clinical situation group of patients with a confirmed COVID-19 infection than without. Working on standardized emergency management of COVID-19 patients at a European level could be useful for future research and would allow relevant reactivity when facing pandemics in the future.

## Figures and Tables

**Figure 1 jpm-12-02085-f001:**
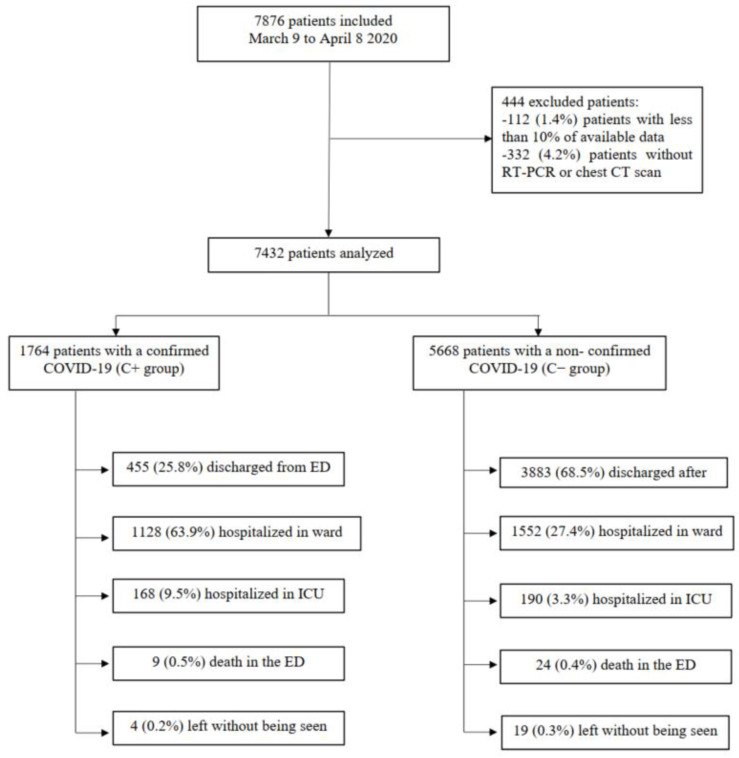
Flow chart of patients included in the study.

**Table 1 jpm-12-02085-t001:** Details of patients included.

		C+ Group (*n* = 1764)				C− Group (*n* = 5668)				
		RT-PCR		Chest CT Scan		RT-PCR		Chest CT Scan		
	Total C+ *n* (%)	Performed *n* (%	Positive *n* (%)	Performed *n* (%)	In Favor *n* (%)	Performed *n* (%)	Positive *n* (%)	Performed *n* (%)	In Favor *n* (%)	Total C− *n* (%)
**TOTAL**	1764 (23.7)	1645 (93.3)	1359 (82.6)	1192 (67.6)	1120 (63.5)	4262 (75.2)	0 (0)	1406 (24.8)	0 (0)	5668 (76.3)

**Table 2 jpm-12-02085-t002:** Characteristics of patients included in the analysis (*n* = 7432). (Description of missing data by variable in Appendix B).

	C+ Group (*n* = 1764)	C− Group (*n* = 5668)	*p*-Value	OR CI 95%
**Demographic data, *n* (%)**				
Mean age in years +/− SD	63.8 +/− 17.5	51.8 +/− 21.1	<0.01	1.86 [1.44; 2.28]
Male	993 (54.6)	2615 (46.1)	<0.01	1.51 [1.36; 1.68]
Current Pregnancy (less 40 yo)	13/96 (13.5)	51/1176 (4.3)	<0.01	3.45 [1.80; 6.60]
**Medical History, *n* (%)**				
Diabetes mellitus all types	373 (21.7)	642 (11.7)	<0.01	2.08 [1.81; 2.40]
Arterial hypertension	755 (43.5)	1287 (22.7)	<0.01	2.62 [2.34; 2.93]
Overweight or obesity	287 (17.5)	515 (23.2)	<0.01	1.68 [1.44; 1.97]
Chronic heart failure	229 (13.0)	520 (9.2)	<0.01	1.48 [1.25; 1.75]
Coronary artery disease	204 (11.6)	401 (7.1)	<0.01	1.73 [1.45; 2.07]
Chronic obstructive pulmonary disease	155 (8.8)	408 (7.2)	<0.01	1.25 [1.03; 1.52]
Asthma	152 (8.6)	641 (11.3)	<0.01	0.74 [0.61; 0.89]
History of Stroke	131 (7.5)	232 (4.1)	<0.01	1.88 [1.51; 2.35]
Active malignant neoplasm	102 (5.8)	317 (5.6)	0.02	1.04 [0.83; 1.31]
Chronic kidney disease	108 (2.3)	235 (4.1)	<0.01	1.51 [1.19; 1.91]
Chronic liver disease	32 (1.8)	73 (1.3)	0.17	1.42 [0.93; 2.16]
Factor of immunosuppression	100 (5.8)	370 (6.8)	0.21	0.84 [0.67; 1.05]
**Habitus, *n* (%)**				
Current smoking	173 (11.2)	1033 (21.2)	<0.01	0.47 [0.40; 0.56]
Alcohol chronic consumption (confirmed or suspected)	68 (4.2)	311 (6.5)	0.04	0.64 [0.49; 0.84]
**Chronic treatment, *n* (%)**				
Angiotensin converting enzyme inhibitors	250 (16.5)	429 (7.6)	<0.01	2.12 [1.79; 2.50]
Angiotensin II receptor blockers	179 (10.6)	288 (5.1)	<0.01	2.21 [1.82; 2.69]
Non-steroidal anti-inflammatory	147 (9.1)	373 (6.6)	0.01	1.30 [1.07; 1.59]

**Table 3 jpm-12-02085-t003:** Description of COVID-19 symptoms and possible mode of contamination. (Description of missing data by variable in Appendix B).

	C+ Group (*n* = 1764)	C− Group (*n* = 5668)	*p*-Value	OR CI 95%
**Mean duration of symptoms +/− SD (in days)**	6.6 +/− 4.7	5.4 +/− 5.6	<0.01	1.17 [1.08; 1.26]
**Possible mode of contamination, *n* (%)**				
Healthcare worker	85 (4.8)	260 (4.6)	<0.01	1.06 [0.82; 1.36]
Institutional living	161 (9.1)	314 (5.5)	<0.01	1.65 [1.35; 2.01]
Notion of COVID-19 contact	470 (32.0)	674 (12.0)	<0.01	3.44 [3.0; 3.44]
**Symptoms, *n* (%)**				
Self-reported feverishness	1214 (68.9)	2335 (41.2)	<0.01	3.15 [2.81; 3.53]
Cough	1175 (67.4)	3378 (61.4)	<0.01	1.30 [1.16; 1.46]
Shortness breath	973 (55.3)	2490 (44.0)	<0.01	1.58 [1.42; 1.76]
Muscle aches	386 (22.8)	1198 (23.3)	0.69	0.98 [0.86; 1.12]
Diarrhea	363 (21.2)	923 (17.8)	<0.01	1.24 [1.08; 1.42]
Headache	273 (15.6)	864 (15.3)	0.79	1.02 [0.88; 1.18]
Chest pain	248 (14.1)	1383 (24.4)	<0.01	0.51 [0.44; 0.59]
Sputum production	215 (12.3)	626 (11.1)	0.17	1.12 [0.95; 1.32]
Ageusia	190 (11.4)	242 (4.9)	<0.01	2.51 [2.06; 3.06]
Vomiting Nausea	180 (10.2)	563 (9.9)	0.72	1.03 [0.86; 1.23]
Abdominal pain	152 (9.0)	448 (8.9)	0.89	1.02 [0.84; 1.24]
Sore throat	148 (8.8)	630 (12.3)	<0.01	0.69 [0.57; 0.83]
Anosmia	153 (8.8)	163 (2.9)	<0.01	3.23 [2.57; 4.06]
Altered consciousness confusion	153 (8.7)	307 (5.4)	<0.01	1.67 [1.36; 2.04]
Runny nose	124 (7.1)	531 (9.4)	<0.01	0.73 [0.60; 0.89]
Agnosia	24 (1.4)	19 (0.3)	<0.01	4.03 [2.20; 7.37]
Skin rash	10 (0.6)	35 (0.6)	0.83	0.93 [0.46; 1.88]

**Table 4 jpm-12-02085-t004:** Vital parameters and clinical examination at admission. (Description of missing data by variable in Appendix B).

	C+ Group (*n* = 1764)	C− Group (*n* = 5668)	*p*-Value	OR CI 95%
**Vital parameter at admission**				
Temperature (in Celsius degree)	37.5 +/− 2.6	37.1 +/− 2.8	0.3	0.94 [0.86; 1.02]
Over 38.5 °C	303 (17.3)	427 (7.8)	<0.01	
Mean heart rate +/− SD	90 +/− 23	91 +/− 24	0.22	1.07 [0.93; 1.21]
Tachycardia (more than 90/min)	821 (47.3)	2602 (49.1)	0.19	0.93 [0.83; 1.04]
Mean respiratory rate +/− SD *	23 +/− 7	21 +/− 6	<0.01	
Over 30 cycles/min *	267 (17.3)	431 (10.5)	<0.01	
Mean systolic blood pressure	133 +/− 35	137 +/− 37	<0.01	0.94 [0.89; 0.99]
Mean diastolic blood pressure	75 +/− 21	79 +/− 22	<0.01	0.83 [0.78; 0.88]
TAS < 90 mmHg	16 (0.9)	56 (1.1)		
Oxygen saturation in room air	94 +/− 2 (*n* = 1392)	97 +/− 3 (*n* = 4888)	<0.01	0.83 [0.77; 0.89]
Oxygen saturation < 90%	248 (17.8)	162 (3.3)	<0.01	
Mental Status *				
GCS 14/15	1506 (95.9)	4242 (97.9)		
GCS 9/13	53 (3.3)	68 (1.6)		
GCS < 9	11 (0.8)	21 (0.5)		
**Clinical examination at admission**				
**Pulmonary auscultation**				
Crackles	998 (57.2)	854 (15.3)	<0.01	7.35 [6.52; 8.27]
Normal	669 (38.4)	4341 (77.7)	<0.01	0.19 [0.17; 0.21]
Other	77 (4.4)	388 (6.9)	<0.01	0.62 [0.48; 0.80]
**Signs of respiratory struggle**				
Swinging thoracoabdominal	116 (6.7)	131 (2.3)	<0.01	3.01 [2.33; 3.89]
Supra-clavicular pulling	150 (8.6)	195 (3.4)	<0.01	2.64 [2.12; 3.29]
Subcostal pulling	101 (5.8)	98 (1.7)	<0.01	3.50 [2.64; 4.65]

Qualitative data are expressed by *n* (%), quantitative data by mean + Standard Deviation. GCS: Glasgow coma scale. * too missing data for Odds Ratio calculation

**Table 5 jpm-12-02085-t005:** Additional tests performed at the admission in emergency department.

	C+ Group (*n* = 1764)	C− Group (*n* = 5668)	*p*-Value	OR CI 95%
**Radiological exam *n* (%)**				
Chest radiography	593 (33.7)	2013 (35.5)	0.14	0.92 [0.82; 1.03]
Infiltrate	391 (65.9)	518 (25.7)	<0.01	5.59 [4.59; 6.81]
Pleural effusion	48 (8.0)	816 (40.5)	<0.01	0.13 [0.10; 0.18]
Normal	98 (16.4)	770 (38.3)	<0.01	0.32 [0.25; 0.40]
**Biological test (mean +/− SD)**				
Haemoglobin (g/L)	13.4 +/− 1.2	13.3 +/−1.5	0.8	0.95 [0.85; 1.05]
White blood count (G/L)	8.6 +/− 9.3	10.2 +/− 8.4	<0.01	0.78 [0.66; 0.90]
Haematocrit (%)	39.8 +/− 6.3	40 +/− 6.2	0.31	0.89 [0.69; 1.09]
Platelets (G/L)	218 +/− 95.4	254 +/− 93.8	<0.01	0.92 [0.78; 1.06]
Urea (mmol/L)	7.8 +/− 5.5	6.5 +/− 4.5	<0.01	1.28 [1.14; 1.42]
Creatinine (μmol/L)	73.4 +/− 59.2	82.6 +/− 52.4	<0.01	0.83 [0.69; 0.97]
Lactate (mmol/L)	1.3 +/− 0.83	0.9 +/− 1	<0.01	1.22 [1.14; 1.32]
Procalcitonin (ng/mL)	0.91 +/− 2.6	0.63 +/− 1.9	0.09	
PCT > 1, *n* (%)	34 (13.9)	60 (10.1)	0.12	
CRP (mg/L)	86.2 +/− 80.4	45.4 +/− 70.4	<0.01	1.86 [1.56; 2.16]
LDH (U/L)	308 +/− 139	253 +/− 99	<0.01	1.65 [1.44; 1.86]
D-dimer (μg/L)	1072 +/− 1057	756 +/− 968	<0.01	1.47 [1.17; 1.77]
Ferritin (ng/mL)	544 +/− 278	263 +/− 259	<0.01	1.95 [1.67; 2.23]

Too missing data for Odds Ratio calculation.

**Table 6 jpm-12-02085-t006:** Therapeutic strategy in emergency department.

	C+ Group (*n* = 1764) *n* (%)	C− Group (*n* = 5668) *n* (%)	*p*-Value	OR IC 95%
**Oxygen therapy in ED**	N = 998	N = 877	<0.01	7.12 [6.32; 8.02]
02 flow	813 (81.4)	779 (88.8)		
1–5 L	535 (65.8)	644 (82.7)		
6–10 L	154 (18.9)	93 (11.9)		
11–15 L	73 (9.0)	17 (2.2)		
>15 L	51 (6.3)	25 (3.2)		
Non-invasive ventilation	142 (14.2)	47 (5.4)	<0.01	10.47 [7.49; 14.63]
Invasive ventilation	43 (4.4)	51 (5.8)	<0.01	2.75 [1.83; 4.14]
Inotropes vasopressors	21 (1.2)	10 (0.2)	<0.01	6.82 [3.21; 14.51]
Antivirals	39 (2.2)	32 (0.6)	<0.01	3.98 [2.49; 6.37]
Antibiotics	350 (20.0)	896 (15.8)	<0.01	1.32 [1.15; 1.51]

**Table 7 jpm-12-02085-t007:** Patient outcome after ED management.

	C+ Group (*n* = 1764) *n* (%)	C− Group (*n* = 5668) *n* (%)	*p*-Value	OR IC 95%
**Outcomes after ED**				
Discharge at home	455 (25.8)	3883 (68.5)	<0.01	0.16 [0.14; 0.18]
Death	9 (0.5)	24 (0.4)	0.68	1.2 [0.56; 2.59]
Left without being seen	4 (0.2)	19 (0.3)	0.63	0.68 [0.23; 2.00]
Hospitalization in ward	1128 (63.9)	1552 (27.4)	<0.01	4.7 [4.2; 5.26]
ICU from ward	154/1128 (13.1)	220/1552 (14.1)	0.46	0.84 [0.67; 1.05]
ICU from ED	168 (9.5)	190 (3.3)	<0.01	3.03 [2.44; 3.76]
**30 Days outcome after ED discharge**				
New ED visit	72 (15.8)	324 (8.3)	<0.01	0.7 [0.54; 0.91]
New hospitalization	56 (12.3)	103 (2.7)	<0.01	1.77 [1.27; 2.46]
Death from all cause at 30 days	241/1702 (14.2)	155/5558 (2.8)	<0.01	5.75 [4.66; 7.09]

## Data Availability

The data presented in this study are available on request from the corresponding author. The data are not publicly available due to local legislation.

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
