# Peer review of "Clinical Characteristics and Management of Patients with a Suspected COVID-19 Infection in Emergency Departments: A European Retrospective Multicenter Study"

_jpm, 2022, doi:10.3390/jpm12122085_

Round 1

Reviewer 1 Report

The authors present an article dealing with the profile and outcome of patients attending ED with either confirmed (C+) or suspected (C-) COVID-19 diagnosis through a multicenter study conducted in Europe. I have some comments:

1.     Data source is quite “old” as the period you consider refers to the first wave in 2020. Did you perform any comparison with latest data? The experience of hcw and diagnostic systems have somehow evolved in almost 2 years. If not this aspect has to be stated as a limitation of the study at least.

2.     It’s not clear what do you mean with “data obtained from the electronic health system”. Indeed every health system is different and data can be obtained either from administrative registers or from ED reports and hospital discharge forms. Please clarify this aspect. 

3.     No definition of “clinical signs suggestive of COVID-19” is provided. Is well known that COVID-19 can present with a wide range of symptoms that can mimic many other conditions. Please specify also in the methods section which vital parameters did you consider.

4.     In objectives you state that “The primary objective of this study was to compare the epidemiology and ED management of patients visiting the ED with suspected COVID-19 infection in a European patient population”, but you didn’t perform any comparison between the various centers, please reformultate the sentence according to results.

5.     Many articles have dealt with risk stratification of COVID-19 patients. How do you contribute towards a better definition of enstablished paths in the management of COVID-19 patients in ED?

Author Response

Dear Editors,

Dear Reviewers, thank you for all yours comments which greatly improve the quality of our manuscript. Our manuscript has been proofread in English via the publisher's system.

The authors present an article dealing with the profile and outcome of patients attending ED with either confirmed (C+) or suspected (C-) COVID-19 diagnosis through a multicenter study conducted in Europe. I have some comments:

  1. Data source is quite “old” as the period you consider refers to the first wave in 2020. Did you perform any comparison with latest data? The experience of how and diagnostic systems have somehow evolved in almost 2 years. If not this aspect has to be stated as a limitation of the study at least.

Answer: Thank you for your relevant comment. In this primary study we only analyzed the first wave in 2020. The comparison with latest data will be performed in future manuscripts. We agree that this limitation must be reported in the manuscript. We added this in the limitation section of the manuscript.

  1. It’s not clear what do you mean with “data obtained from the electronic health system”. Indeed every health system is different and data can be obtained either from administrative registers or from ED reports and hospital discharge forms. Please clarify this aspect. 

Answer: We agree that the methods must be clarified. The data was obtained from the patient chart review of each center by local study investigators. Local investigator reviewed the patient chart of ED but also hospitalization report if necessary to evaluate if the patient died during the hospitalization.

About the 30-days status following the ED consultation, local investigators only checked in the local patient chart review if the patient has re-visited to the ED, hospitalization or death was reported.

We modified the manuscript accordingly.

  1. No definition of “clinical signs suggestive of COVID-19” is provided. Is well known that COVID-19 can present with a wide range of symptoms that can mimic many other conditions. Please specify also in the methods section which vital parameters did you consider.

Answer: Thank you for your comment. We agree that the reader must have the details of all symptoms that may suggest COVID-19. We agree that a wide range of symptoms could evocate COVID-19. The symptoms considered by the authors are listed in table 3. In order not to make the presentation too heavy, we suggest adding table 3 as references.

Now we specify in the text the vital parameters considered.

  1. In objectives you state that “The primary objective of this study was to compare the epidemiology and ED management of patients visiting the ED with suspected COVID-19 infection in a European patient population”, but you didn’t perform any comparison between the various centers, please reformultate the sentence according to results.

Answer: We agree that the objectives must be reformulated. The objectives were to describe and compare the profile and outcome of patients attending the ED with a confirmed COVID-19 infection to patients with a suspected COVID-19 but tested negative for COVID-19. We modified the manuscript accordingly.

  1. Many articles have dealt with risk stratification of COVID-19 patients. How do you contribute towards a better definition of enstablished paths in the management of COVID-19 patients in ED?

Answer: Our manuscript is a primary one and only descrive this population in European EDs. We are currently finalizing the analysis of risk stratification of COVID-19 patient that will be included in a planned secondary analysis of our data.

Reviewer 2 Report

Dear authors,

thank you for the possibility to review this interesting paper.

These are my comments:

45/46 and 287/288 „We showed overall a more severe group with a confirmed COVID-19 infection.“ -> What does this mean? „more severely affected“, „more severely symptomatic“? or „more critical clinical situation“ (as stated in l. 170)

Can you tell whether the C – group had other confirmed viral illnesses, such as influenza or RSV? Otherwise the C- group is a rather heterogenous group which should be  stated as a limitation of the study.

l. 85 ff. Who decided whether a patient was considered „suspicion of COVID-19“? The patient? Admitting stuff? Emergency physician?

Were patients included in the study who presented with non-typical symptoms for COVID-19 (at first no suspicion of COVID-19) and nonetheless tested positive?

l. 90 „at the initial consultation in the ED“ -> sometimes PCR-tests become positive later in the course, but as I understand this study, this was not taken into account. Should be mentioned as a limitation.

Figure 1: 3883 (68.5%) disharged after-> means d/c from ED?

You state that you have excluded 332 patients without RT-PCR or chest CT-scan (ll. 126/127). But in table 1 ("patients included") you state that 3860 patients in the C- group had a RT-PCR and 1220 had a chest CT scan performed. These are altogether 5080 patients. However, the total number is 5668 which would mean that almost 600 included patients of the C- group did not receive a PCR or chest CT scan. But this was stated as a criterion of exclusion.

153 patient -> patients

202 no patients were lost in follow-up?

233/234 „39% of patients in the C- group didn´t benefit from a RT-PCR in the ED“ -> What does that mean? Which patient of the C- group did benefit from a RT-PCR and why?

241 How often did CT scan w/ contrast change management, e.g. because pulmonary embolism was diagnosed

254 did patients of the C- group return and were then tested positive?

282 But you did collect results of RT-PCR or chest CT scan if performed later in the course if patients were admitted?

Can you please use the same spelling for „COVID-19“:

in l. 287 „Covid19 infection“,  in 284 „Covid-19 patients“, ll. 275/276 „COVID“ and most of the time „COVID-19“

321/322 is this one person with different spelling working in two institutions?

Appendix 1:

What does the category „Male“ mean? Is this the category sex (male/female/div.)?

Category: „Diabetes“ means „Diabetes mellitus all types“?

 „Men duration of symptoms in days“ -> you probably meant „Mean duration….“

Body mass index: No number included in C+ group

Are you sure that you are not missing data on whether the temperature was over 38.5, but you are indeed missing the temperature itself in 16 and 238 cases (C- / C+ group)?

ll. 335 ff.: is this information for the author group? Should these 2 references be added?

I hope these comments can be helpful. I am willing to review again.

Kind regards

Author Response

These are my comments:

  1. 45/46 and 287/288 „We showed overall a more severe group with a confirmed COVID-19 infection.“ -> What does this mean? „more severely affected“, „more severely symptomatic“? or „more critical clinical situation“ (as stated in l. 170)

Answer: Thank you for this comment. "More severely affected" means "more critical clinical situation". We modified the manuscript accordingly.

  1. Can you tell whether the C – group had other confirmed viral illnesses, such as influenza or RSV? Otherwise the C- group is a rather heterogenous group which should be  stated as a limitation of the study.

Answer: Patients who were categorized as C- may indeed have had another viral or infectious disease. In case of viral involvement, patients did not benefit from multiplex viral testing. They could therefore have had RSV or influenza but this was not investigated as it would not have changed their ED management.

  1. 85 ff. Who decided whether a patient was considered „suspicion of COVID-19“? The patient? Admitting stuff? Emergency physician?

Answer: The organization of the emergency departments that participated to our study in Europe may have been different. Indeed, the reception and triage of patients could be done either by an emergency doctor or by a triage nurse. This standard organization was not modified during the COVID-19 pandemic.

  1. Were patients included in the study who presented with non-typical symptoms for COVID-19 (at first no suspicion of COVID-19) and nonetheless tested positive?

Answer: We agree that this point needs to be clarified. We only included patients with a suspected COVID-19 infection. Patients for whom COVID-19 was discovered incidentally (asymptomatic patients) were not included in the study. We clarified this point in the manuscript.

  1. 90 „at the initial consultation in the ED“ -> sometimes PCR-tests become positive later in the course, but as I understand this study, this was not taken into account. Should be mentioned as a limitation.

Answer: Thanks for this relevant remark. We reported this point in the limitation section.

“Second, we did not prospectively follow-up included patients and we did not collect the results of RT-PCRs performed after ED discharge. Therefore, some patients could be false negatives with a PCR that would be positive at a later stage.”

  1. Figure 1: 3883 (68.5%) disharged after-> means d/c from ED?

Answer: The formatting of the flow chart has cut off the sentence: "3 883 (68.5%) discharged after ED". We modified the flow chart accordingly.

Figure 1. Flow chart of patients included in the study.

  1. You state that you have excluded 332 patients without RT-PCR or chest CT-scan (ll. 126/127). But in table 1 ("patients included") you state that 3860 patients in the C- group had a RT-PCR and 1220 had a chest CT scan performed. These are altogether 5080 patients. However, the total number is 5668 which would mean that almost 600 included patients of the C- group did not receive a PCR or chest CT scan. But this was stated as a criterion of exclusion.

Answer: Thank you very much for this point. This was a reporting error. We modified the Table 1 and the manuscript accordingly.

Table 1. Details of patients included  

    C+ group (n=1 764) C- group (n=5 668)  
    RT- PCR Chest CT scan RT- PCR Chest CT scan  
  Total C+n (%) Performedn(% Positiven(%) Performedn(%) In favorn(%) Performedn(%) Positiven(%) Performedn(%) In favorn(%) Total C-n(%)
TOTAL 1 764 (23.7) 1645 (93.3) 1359 (82.6) 1192 (67.6) 1120 (63.5) 4262 (75.2) 0 (0) 1406 (24.8) 0 (0) 5 668 (76.3)

  1. 153 patient -> patients

Answer: Sorry for the spelling mistake. We have changed the sentence “Moreover, patients in the C+ group had more comorbidities than patients in the C- group (i.e., diabetes mellitus all types, arterial hypertension, chronic heart failure, coronary artery disease or stroke)”.

  1. 202 no patients were lost in follow-up?

Answer: We agree that this point need to be clarified. The data was obtained from the patient chart review of each center by local study investigators. Local investigator reviewed the patient chart of ED but also hospitalization report if necessary to evaluate if the patient died during the hospitalization.

About the 30-days status following the ED consultation, local investigators only check in the local electronic health system if the patient has re-visited, hospitalized or death was reported. Neither the reason for reconsultation nor the clinical or biological data were recorded during this reconsultation. There was no follow-up with recall of patients or relatives.

We modified the manuscript accordingly.

  1. 233/234 „39% of patients in the C- group didn´t benefit from a RT-PCR in the ED“ -> What does that mean? Which patient of the C- group did benefit from a RT-PCR and why?

Answer: Sorry for this mistake. This sentence does not fit the context of our study. We have deleted this sentence.

  1. 241 How often did CT scan w/ contrast change management, e.g. because pulmonary embolism was diagnosed

Answer:  Thanks for this remark. We did not record this data. The injection of contrast material was a real debate during the pandemic. As the protocols were all different from one centre to another, we did not record this data.

  1. 254 did patients of the C- group return and were then tested positive?

Answer:  As stated above, About the 30-days status following the ED consultation, local investigators only check in the local electronic health system if the patient has re-visited, hospitalized or death was reported. Neither the reason for reconsultation nor the clinical or biological data were recorded during this reconsultation.There was no follow-up with recall of patients or relatives.

We modified the manuscript accordingly.

  1. 282 But you did collect results of RT-PCR or chest CT scan if performed later in the course if patients were admitted?

Answer : As stated above, the data was obtained from the patient chart review of each center by local study investigators. Local investigator reviewed the patient chart of ED but also hospitalization report if necessary to evaluate if the patient died during the hospitalization.

About the 30-days status following the ED consultation, local investigators only check in the local electronic health system if the patient has re-visited, hospitalized or death was reported.

We did not collect results of RT-PCR or chest CT scan if it was performed later or controlled after ED consultation. In our study we wanted to keep an overview of the situation in the emergency department. We modified the manuscript accordingly.

  1. Can you please use the same spelling for „COVID-19“. In l. 287 „Covid19 infection“,  in 284 „Covid-19 patients“, ll. 275/276 „COVID“ and most of the time „COVID-19

Answer:  We agree that the spelling must be standardized. We modified the manuscript with the term “COVID-19”.

  1. 321/322 is this one person with different spelling working in two institutions?

Answer: Sorry for this mistake. Dr Möckel currently work in the Charité (Berlin). We modified the manuscript accordingly.

Appendix 1:

  1. What does the category „Male“ mean? Is this the category sex (male/female/div.)?

Answer: we agree that this title should be changed. We reported sex rather than male.

  1. Category: „Diabetes“ means „Diabetes mellitus all types“?

Answer: We agree that this must be clarified. Diabetes means diabetes mellitus all types. We modified the manuscript accordingly.

  1. “Men duration of symptoms in days” -> you probably meant „Mean duration….“

Answer: Thanks for this remark. We modified the manuscript accordingly.

  1. Body mass index: No number included in C+ group

Answer: We added this data.

  1. Are you sure that you are not missing data on whether the temperature was over 38.5, but you are indeed missing the temperature itself in 16 and 238 cases (C- / C+ group)?

Answer: Indeed, this remark is relevant. We initially expressed the data “temperature was over 38.5” among patients with known temperature data. We agree we must modified the appendix material.

  1. 335 ff.: is this information for the author group? Should these 2 references be added?

Answer: Sorry, we have removed this information which was intended for discussion between the authors

Round 2

Reviewer 1 Report

The authors have clarified several of the questions I raised in my previous review. Unfortunately, the article has some fundamental flaws in the experimental design and, most critically, with the analysis.

First of all, COVID-19 is diagnosed by direct detection of SARS-CoV-2 RNA using nucleic acid amplification tests (NAATs) or by detection of viral protein using an antigen test. A positive NAAT or antigen test is generally indicative of infection and does not need to be repeated. A chest CT or x-ray cannot accurately distinguish between COVID-19 and other respiratory infections, like seasonal flu. Therefore, people who only had a CT scan cannot be included in the C+ group.

Moreover, the analysis is incomplete: the authors didn’t present an odds ratio analysis nor studied any confounding factor.

Author Response

Peer-reviewers 1, Round 2:

The authors have clarified several of the questions I raised in my previous review. Unfortunately, the article has some fundamental flaws in the experimental design and, most critically, with the analysis.

First of all, COVID-19 is diagnosed by direct detection of SARS-CoV-2 RNA using nucleic acid amplification tests (NAATs) or by detection of viral protein using an antigen test. A positive NAAT or antigen test is generally indicative of infection and does not need to be repeated. A chest CT or x-ray cannot accurately distinguish between COVID-19 and other respiratory infections, like seasonal flu. Therefore, people who only had a CT scan cannot be included in the C+ group.

Moreover, the analysis is incomplete: the authors didn’t present an odds ratio analysis nor studied any confounding factor.

Answer: Thank you very much for your remarks.  The aim of our manuscript was to describe the management of patients with COVID-19. During the first wave, RT-PCR and antigenic test were not available in most EDs. And, in EDs where the RT-PCR was available, the delay was too huge to use this as the sole triage tool [1]. The CT-scan was a relevant and complementary screening tool to RT-PCR. During the first wave, patients with a positive CT-scan were managed similarly to patients with a positive RT-PCR. Therefore we are willing to keep patients with a positive CT-scan in the C+ group. We added the following references showing that a positive CT-scan is associated with a COVID-19 infection [2].

Regarding your second point, we completed the analysis with odds ratio. We modified the statistical section accordingly. Please, find above all updated tables.

[1] Cancella De Abreu, Chauvin A, Peyrony O, et al. Time-to result advantage of Point-Of-Care SARS-CoV-2 PCR testing to confirm COVID-19 in Emergency Department: a retrospective multicenter study. Eur J Emerg Med. In Press

[2] De Smet K, De Smet D, Ryckaert T, et al. Diagnostic Performance of Chest CT for SARS-CoV-2 Infection in Individuals with or without COVID-19 Symptoms. Radiology. 2021 Jan;298(1):E30-E37.

Round 3

Reviewer 1 Report

The writers addressed my suggestion and modified the paper accordingly. I appreciated that the authors implemented the statistical analysis that was previously missing. Despite the rather small temporal and spatial extent of the study, I think that the paper is clear and well-written and give a good insight into differences between COVID-19-positive patients and suspected COVID-19 patients during the first pandemic wave. Overall I think your article should be published